# Challenges in the medical oxygen ecosystem of Peru: A political economy analysis

Patricia J. Garcia[1]*, Freddy Eric Kitutu[2,3,4], Jehnette M. Guzman[1], Lizzete Najarro[1], Freddie Ssengooba[4], Carina King[5]

1 School of Public Health, Universidad Peruana Cayetano Heredia, Lima, Peru, 2 Department of Pharmacy, Makerere University School of Health Sciences, Kampala, Uganda, 3 Department of Women's and Children's Health, International Child Health and Migration, Uppsala University, Uppsala, Sweden, 4 Makerere University School of Public Health, Kampala, Uganda, 5 Department of Global Public Health, Karolinska Institutet, Stockholm, Sweden

* pattyjannet@gmail.com, patricia.garcia@upch.pe

## Abstract

Medical oxygen is essential in the management of several human disease conditions including acute respiratory conditions across the life course, and yet access remains unequal in many low- and middle-income countries, including Peru. This study explores Peru's challenges in ensuring reliable oxygen supplies, with a focus on those laid bare or exacerbated by the COVID-19 pandemic, to inform strategies for improving medical oxygen access. Using a political economy analysis, we conducted 13 key informant interviews with stakeholders involved in oxygen policy, supply, and health care, supported by reviews of 117 academic and grey literature sources, including policy documents. Before the COVID-19 pandemic, Peru's requirement for medical oxygen to be > 99% pure restricted competition, consolidating control of a few large liquid oxygen suppliers on the oxygen market and blocking smaller, affordable providers due to high compliance costs. Although pre-pandemic oxygen supplies were reportedly adequate, the pandemic exposed severe limitations including market constraints, slow government response, and lack of data management, resulting in an acute oxygen crisis. Civil society and private organizations stepped in, donating medical oxygen generator plants, but many of these are now unused due to insufficient planning for maintenance and operation. This study underscores the urgent need for a National Oxygen System in Peru to oversee supply, distribution, and maintenance, and strengthen resilience for future health emergencies. Solutions include reducing reliance on a small number of external suppliers, infrastructure investments, dedicated funding for maintenance, and training for personnel to ensure continuous oxygen access nationwide. This research highlights systemic vulnerabilities in Peru's health system and calls for coordinated policies to ensure equitable oxygen access and preparedness for future crises.

**Data availability statement:** All anonymized transcripts (DATA) are included in S6 Appendix.

**Funding:** This work was supported by the US Agency for International Development (USAID) under contract no. 7200AA19C00002 to FEK. Cayetano Heredia University (Lima, Peru) provided salary support to PJG, JMG, and LN; Makerere University provided salary support to FEK and FS; and the Karolinska Institute provided salary support to CK. The funders had no role in study design, data collection and analysis, decision to publish, or preparation of the manuscript.

**Competing interests:** The authors declared that they have no competing interests exist.

## Introduction

Medical oxygen is a critical yet often overlooked medicine, essential for treating a range of conditions, including respiratory illnesses and critical conditions. It has been in clinical use since the late 1800s [1]. In 2002, the World Health Organization (WHO) included oxygen in the essential medicine list (EML) – but it was limited to anesthetic use, and only in 2017 was the indication to be used for hypoxemia added [2].

A large and diverse patient population relies on oxygen therapy, spanning from vulnerable newborns in respiratory distress to children with lower respiratory tract infections like pneumonia or bronchiolitis, sepsis, and congenital or acquired heart disease [3]. Adults also face critical oxygen needs due to infectious causes of hypoxemia (such as tuberculosis, malaria, pneumonia, and HIV/AIDS), chronic respiratory illnesses (like COPD, asthma, pulmonary hypertension, and interstitial lung disease), and other hypoxemic conditions including ARDS, trauma, heart failure, post-surgical recovery, resuscitated sepsis, and cancer [4]. The onset of the COVID-19 pandemic led to a substantial increase in the demand for oxygen due to the respiratory complications associated with the virus and highlighted the severe gaps and inequities in oxygen supply across health systems globally [5,6]. The acute oxygen shortages experienced during the COVID-19 pandemic emerged as a critical disparity, underscoring the lack of preparedness and security in essential healthcare resources worldwide [7–9].

Peru is a Latin American (LA) middle-income country of roughly 34 million inhabitants of which around 2.9 million are under five years of age [10]. In the two decades preceding the COVID-19 pandemic, Peru experienced steady economic growth and expanded healthcare access, yet significant socio-economic, regional, and ethnic disparities in healthcare access persisted [11–13]. Life expectancy saw a significant increase by 2019, rising to 81.8 years for women and 78.7 years for men, compared to 71.6 and 67 years in 1990. This improvement is largely attributed to advancements in maternal and child healthcare and a substantial decline in mortality from infectious diseases [14].

The surge in COVID-19 cases in the country stressed the national healthcare system, revealing a lack of preparedness in oxygen availability and distribution affecting the management of cases and mortality. According to data from the Global Burden of Disease study, Peru experienced the highest excess mortality rate due to COVID-19 in 2020, with 413.4 excess deaths per 100,000 population (range: 410.3–416.1). As a result, life expectancy in 2021 regressed closer to 1990 levels, falling to 74.9 years for women and 68.8 years for men [15].

This paper examines the structural and policy-related obstacles Peru faced in ensuring adequate access to medical oxygen for those who need it, providing key insights for the global political economy of oxygen presented in the Lancet Global Health Commission on Medical Oxygen Security [16].

## Methods

### Study design

We used a problem-driven political economy analysis (PEA) to examine the structural and political constraints shaping Peru's medical oxygen ecosystem. This approach

focuses on a specific problem—in this case, the acute oxygen shortages during COVID-19—and explores the underlying power dynamics, institutional arrangements, and incentives that sustain it. Unlike generic sectoral assessments, problem-based PEA help explain not just what is happening, but why, by identifying stakeholders' interests, barriers to reform, and potential entry points for change [17, 18].

We conducted a PEA using qualitative data from key informant interviews and a synthesis of academic literature, grey literature and health system policy documents. The analysis explored the intersection of regulatory frameworks, stakeholder dynamics, and economic factors shaping Peru's medical oxygen system. This study builds on previous political economy work in subnational health systems and served as one of the country case studies informing the *Lancet Global Health Commission on Medical Oxygen Security*, with a specific focus on the political context in Peru. [19–21]

### Setting

The country is highly centralized politically and economically in its capital city Lima, which accounts for around 1/3 of the total population, and 46% of the national gross domestic product (GDP) [22]. Similarly, to other LA countries, Peru is considered as predominantly urban, with 20.7% of the population falling under the rural category [23]. Data from 2019 from show that lower respiratory tract infections (including tuberculosis) remained a leading cause of death in the country, accounting for 13.2% of all deaths. Cancer (21.9%) and cardiovascular diseases (19.2%) were the two most common causes overall. Other significant contributors included chronic kidney disease and diabetes (collectively accounting for 8.4%) and chronic respiratory conditions were responsible for 4.6% of deaths [10].

### Qualitative interviews

For the qualitative component, we identified and invited key informants who had actively participated in the design, financing, regulation, implementation, or advocacy of medical oxygen policies in Peru over the past five years. Participants were purposefully selected based on stakeholder mapping, with additional participants recruited through the snowball sampling method. Most interviews were conducted virtually via Zoom between June 7 and July 25, 2023. Three participants requested in-person interviews, all of which were conducted in a quiet space at a coffee shop. All invited key informants agreed to participate, and no compensation was provided. Interviews were conducted in Spanish by a social scientist experienced in in-depth interviewing, following a structured guide (S1 Appendix). An assistant was also present to take notes. All interviews were recorded and transcribed.

### Document search

The published academic literature search was performed in the following databases: Medline, Embase, Web of Science and Global Index Medicus with the last search conducted on April 03, 2023, using the method described by Bramer et al [24]. The search strategy was developed in Medline (Ovid) in collaboration with librarians at the Karolinska Institutet University Library. For each search concept Medical Subject Headings (MeSH-terms) and free text terms were identified. No language restriction was applied. The search was performed for several countries; we reviewed the papers related to Peru. Full search strategies for all databases are available in S2 Appendix.

Gray literature was sourced using Google with three search terms: "Oxígeno medicinal Perú" (Medical oxygen Peru), "Oxígeno y COVID-19" (Oxygen and COVID-19), and "Planta de oxígeno medicinal Perú" (Medicinal oxygen plant Peru). The first 100 results for each term were reviewed, including documents and videos, while excluding advertisements, policy documents, and regulations.

A policy search was conducted using the Peruvian Ministry of Justice database [25] which provides open access to all national laws, norms, and regulations and features a search engine that allows filtering by date, type of norm, code, public sector, content, or keyword. We did not restrict the search by time frame or public sector, using the keyword "oxígeno medicinal" (medicinal oxygen).

## Analysis

The qualitative interviews were analyzed using the conceptual framework method [26]. Two randomly chosen transcripts were critically read to identify initial themes and categories, considering the content of the text. Subsequently, the interviews were coded, with the aim of looking for similarities and differences in the discourses of the interviewees. For the document search we created a matrix for data abstraction. Each document was reviewed by two independent reviewers who generated short summaries of portions of the text which were introduced into the relevant cell of the matrix. The information from the different sources, qualitative interviews, literature and policies was integrated to create a narrative using the main themes found during the interviews.

## Ethical approval and consent to participate

The research was reviewed and approved by the Cayetano Heredia University IRB (approval number 211379). Written informed consent to participate in the study was obtained from all human research participants.

## Results

A total of 13 interviews were conducted with stakeholders from national and subnational government institutions, civil society, the private sector, healthcare providers, medical professional associations, and multilateral agencies. Most participants were aged 45 and older, approximately one-third were women, and the majority had 20–30 years of professional experience (Table 1)

In the academic literature we found 631 articles, but only 7 were identified as relevant (see Prisma flow in S3 Appendix). The grey literature review resulted in 87 pertinent documents (see S4 Appendix) and in the policy review from 36 policies found, 23 were included in this analysis (S5 Appendix).

## Peru contextual issues

Peru's health system is highly fragmented, with distinct subsystems (public, social security (EsSalud) and private) operating independently under the oversight of the Ministry of Health (MINSA). Additionally, Peru is divided in 24 regions, each with its own government, budget and health directorate (decentralization). This segmentation has limited coordinated responses to public health needs, with medical oxygen as a critical example [27]. One important contextual issue in Peru

**Table 1. Characteristics of participants in the key informant interviews.**

| Code | Sector | Type of institution/Role | Years of experience | Gender |
|------|--------|--------------------------|---------------------|--------|
| 001 | Government | National Regulatory Agency/Directive | +30 y | F |
| 002 | Civil Society | NGO/ in charge of importations & logistics | 20 y | M |
| 003 | Multilateral Agency | Health Cooperation agency/ technical advisor | +20 y | M |
| 004 | Private Sector | Company dedicated to the sale of construction and mining machinery/ involved in Construction of medical oxygen plants during COVID-19 | +30 y | M |
| 005 | Government | Ministry of Health/ Advisor | +30 y | M |
| 006 | Private Sector | Member of the National Society of Industries | +20 y | M |
| 007 | Government | Ministry of Health/ Directive | +10 y | F |
| 008 | Professional Society | Directive from Medical Professional Society | +20 y | M |
| 009 | Civil Society | Catholic Priest/ Regional movement to provide oxygen to patients | 30 y | M |
| 010 | Government | Subnational Regulatory Agency/ Directive | +10 y | F |
| 011 | Government | Ministry of Health/ Infrastructure and Equipment Offices | 30 y | M |
| 012 | Health Provider | Subnational Primary Health Provider | 20 y | F |
| 013 | Health Provider | Subnational Hospital Health Provider | +20 y | M |

has been the political turmoil, in part associated to the corruption scandal in Latin America related to Odebrecht [28]. As a result of the political instability, between 2016–2022, Peru has seen 7 different presidents (lasting on average just 8.5 months and the change of 17 Ministers of Health in that 6-year period, 8 of those during the pandemic years of 2020 and 2021 alone. Peru began its pandemic response preparations early and implemented several public health measures, aware of the weaknesses of its fragmented health system and the political situation [29]. However, wide inequalities, with 70% of the population engaged in informal work with precarious income and social protection, poor housing and over-crowding, and an extended family culture in opposition to social distancing, all added to the perfect storm which resulted in very high rates of infection [30].

Despite MINSA's role in regulating medical oxygen, the lack of a national oxygen strategy meant that each subsystem and each region operated its procurement and distribution channels, leading to inconsistencies and shortages during the COVID-19 crisis. The pandemic highlighted the limitations of the decentralized approach, as rural regions and areas outside Lima experienced severe oxygen shortages. Policies governing medical oxygen were not uniformly applied across regions, reflecting the underlying inequities exacerbated by fragmented governance.

### Regulatory and Economic Barriers to Oxygen Access: The policy trap

Although Peru approved the First National Policy for Medicines in 2004, the first essential medicine´s list (EML) was not approved until 2010 [31]. Before this, oxygen was classified as a medical device with no specific regulations. The 2010 EML introduced oxygen as a medication with a requirement of a concentration standard of >99% [32,33].

Starting in 2011, medical oxygen, like any other medicine, required a **sanitary registration** through DIGEMID (Peru's national authority for medicines, medical devices, and diagnostics). Sanitary registration is the official process or documentation necessary for the legal approval and commercialization of pharmaceutical products in Peru. For medical oxygen, this process enforces standards for manufacturing, distribution, and labeling, including compliance with good manufacturing practices for oxygen production plants [34].

In 2012, INDECOPI (The Peruvian Institute for the Defense of Competition and the Protection of Intellectual Property) declared the >99% concentration requirement an illegal bureaucratic barrier, yet MINSA maintained this standard. The 2018 Ministerial Resolution updated the EML, still requiring >99% purity for medical oxygen. Fig 1 summarizes the main regulations regarding medical oxygen in Peru, before the pandemic.

This led to six companies holding sanitary registrations to commercialize medical oxygen in Peru before the pandemic: Praxair Peru, Linde Gas Peru, Tecnogas, Air Products Peru, Indura Peru, and Oxyman Comercial. These companies are organized into two business groups, producing both industrial oxygen for the mining and metallurgical sectors (their primary clients) and medical oxygen. Together, these groups supplied 92.6% of the medical oxygen contracted by the government from 2008 to 2020, according to the State Contracting Supervisory Body (OSCE as per its Spanish acronym) [35–40]. (Table 2)

Interviewees also noted that pre-pandemic oxygen regulations in Peru were influenced by political and economic interests, requiring a >99% purity level despite the WHO's recommendation of 93% or higher.

> *"…(regarding the policy that changed the level of purity of medical oxygen)…it was a political and economic issue because, by raising the level of oxygen purity it was directed to two companies that were the only ones that did meet that requirement… it was like giving the way only to the two of them because the rest of the companies were not going to comply…"*
>
> *Participant 001*

Peru's regulation requiring medical oxygen purity levels of >99% limited market competition by concentrating control within a few large suppliers, creating a "policy trap" that prevented smaller, more affordable providers from entering the market due to the costly technology needed to meet these standards.

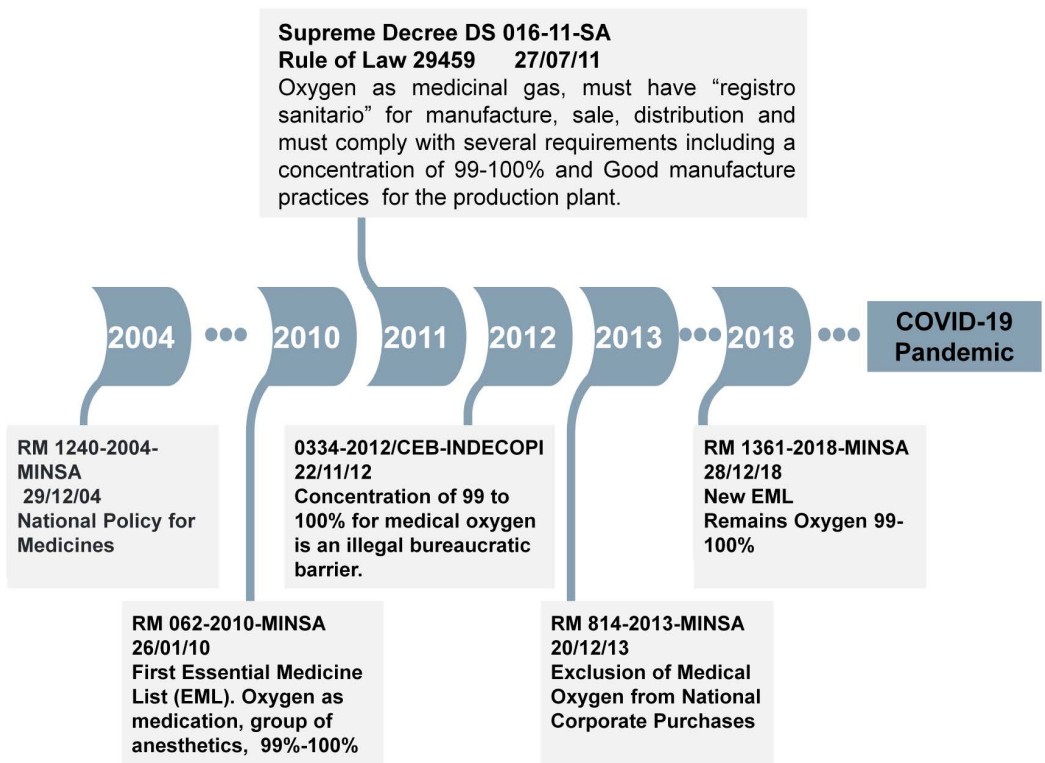

RM= Ministerial Resolution

**Fig 1. Timeline of the Regulations regarding Medical Oxygen in Peru before the COVID-19 Pandemic.**

We found no objective data on pre-pandemic oxygen demand or supply levels, particularly in peripheral hospitals. Interviewees reported no awareness of shortages, but this may reflect limited monitoring and under-recognition of oxygen needs in some settings. Due to varying conditions across HEs nationwide, oxygen distribution within facilities was managed in different ways: some used installed networks or purchased oxygen in cylinders, while others relied solely on refillable cylinders. Large HEs such as hospitals were responsible for procuring and replenishing their oxygen supplies. In smaller HEs, typically part of a health network, oxygen procurement was handled by the network administration or, in regional areas, by the regional government.

*"The supply of oxygen (before COVID) was sufficient, at least I have never been aware of a shortage that has put people's lives at risk"*

*Participant 013*

*"... What we had at the time was enough for the patients who came to the hospital. At least in large hospitals, here in Lima, it seemed to me that we were in the right amount. In the periphery hospitals, in the regional hospitals, there is still no liquid medical oxygen tank (...) an ICU in a region or a province, works with oxygen cylinders and every four or six hours they have to be replaced, it became customary at that time and that's how they have worked."*

*Participant 008*

**Table 2. Main groups of Medical Oxygen in Peru contracting with the government: MINSA, EsSalud and regional governments Period 2008–2020*.**

| Business group | Country | Peruvian company | Contracts (Million USD, %)** | Location of Production Plants *** | Purity level | Production method | Distribution method |
|---|---|---|---|---|---|---|---|
| Linde-Praxair 88.09 M USD 70.8% | Germany | Praxair Peru | 56.23 (45.2%) | Callao Ica Lima | >99% | LOX | Liquid oxygen tanks or Gaseous cylinders |
| | | Linde Gas Peru | 20.93 (16.8%) | | | | |
| | | Tecnogas | 10.93 (8.8%) | | | | |
| Air Products 27.17 M USD 21.8% | USA | Air Products Peru | 17.80 (14.3%) | Chimbote | >99% | LOX | Liquid oxygen tanks or Gaseous cylinders |
| | | Indura Peru | 9.37 (7.5%) | | | | |
| Oxyman Comercial. | Peru | Oxyman Comercial | 4.06 (3.3%) | NA | >99% | Does not produce Buys LOX | Sells oxygen in gaseous cylinders |
| OxyCusco | Peru | OxyCusco | 0.20 (0.2%) | Cusco | >99% | PSA | Gaseous cylinders |
| Oxigeno Iquitos | Peru | Oxigeno Iquitos | 1.0 (0.8%) | Loreto | 93% | PSA | Gaseous cylinders |
| Oxigeno Loreto | Peru | Oxigeno Loreto | 1.59 (1.3%) | Ucayali | >99%, 93%**** | PSA | Gaseous cylinders |
| Oxigeno Tao | Peru | Oxigeno Tao | 0.67 (0.5%) | Loreto | 93% | PSA | Gaseous cylinders |
| Villalobos Rufino Ivan | Peru | Villalobos Rufino Ivan | 1.59 (1.3%) | San Martin | 93% | Sale and distribution of oxygen produce by "Oxigeno San Martin" | Gaseous cylinders |
| Total Million USD | | | 124.37 (100%) | | | | |

Source: Prepared with data for the three first business groups from reference [41] and from the Data base from the State Contracting Supervisory Body (OSCE). LOX = Liquid oxygen

* We included only companies with registro sanitario before the pandemic.

**Using exchange rate from 2008: 0.3124 USD per PEN

*** According to the DIGEMID Sanitary registration

****Exception for regions with difficult access.

## COVID-19 and Oxygen in Peru: limited market, slow response and lack of organization and data resulted in a crisis

In March 2020, Peru declared a mandatory quarantine, yet COVID-19 infections, hospitalizations, and deaths continued to rise. Oxygen became essential for critical COVID-19 patients, creating a major supply challenge for the government.

The Ombudsman's Office reported that at least 20% of hospitalized and ICU patients required oxygen, but supplies were insufficient [35]. Initially, the full impact of COVID-19 as a respiratory illness and the demand for oxygen were underestimated, leading to a crisis as the health system became overwhelmed.

*"... In the months of June, July (2020), when the issue of oxygen was already complicated, we saw people in the streets with their cylinders trying to buy oxygen, and also, we saw people dying because of the lack of oxygen..."*

*Participant 002*

Most high-purity liquid medical oxygen (LMO) plants are located along the central coast of Peru, while the few plants in the Andean and jungle regions are pressure swing adsorption (PSA) plants that produce only limited quantities of gaseous oxygen at 93% purity [40]. High-purity packaging facilities (booster compression system) are also primarily coastal, which led to major disparities in oxygen access. During the pandemic, these inequalities became evident as Andean and jungle regions faced severe shortages—not only due to limited local oxygen production but also because of inadequate

distribution infrastructure. This included a lack of both LMO tankers and vehicles for transporting gaseous oxygen cylinders, as well as poor road conditions in the highlands and the absence of road access to key jungle cities like Iquitos.

In the early months of the pandemic, industrial suppliers initially continued delivering oxygen despite reduced operations. However, as economic activities resumed, companies shifted back to industrial oxygen production and were unable to meet the growing demand for medical oxygen.

In response, the government implemented policies to encourage local production and oxygen imports (Fig 2). For example, an emergency decree that temporarily authorized 93% oxygen purity and included measures to boost production and access. However, smaller companies struggled to obtain import permits or lacked the infrastructure to produce medical oxygen due to an absent market in prior years. Due to the oxygen shortage and rising demand, both legal and illegal small-scale vendors ("black market") began selling oxygen cylinders directly to the public at inflated prices—sometimes using stolen hospital cylinders or equipment not intended for medical use.

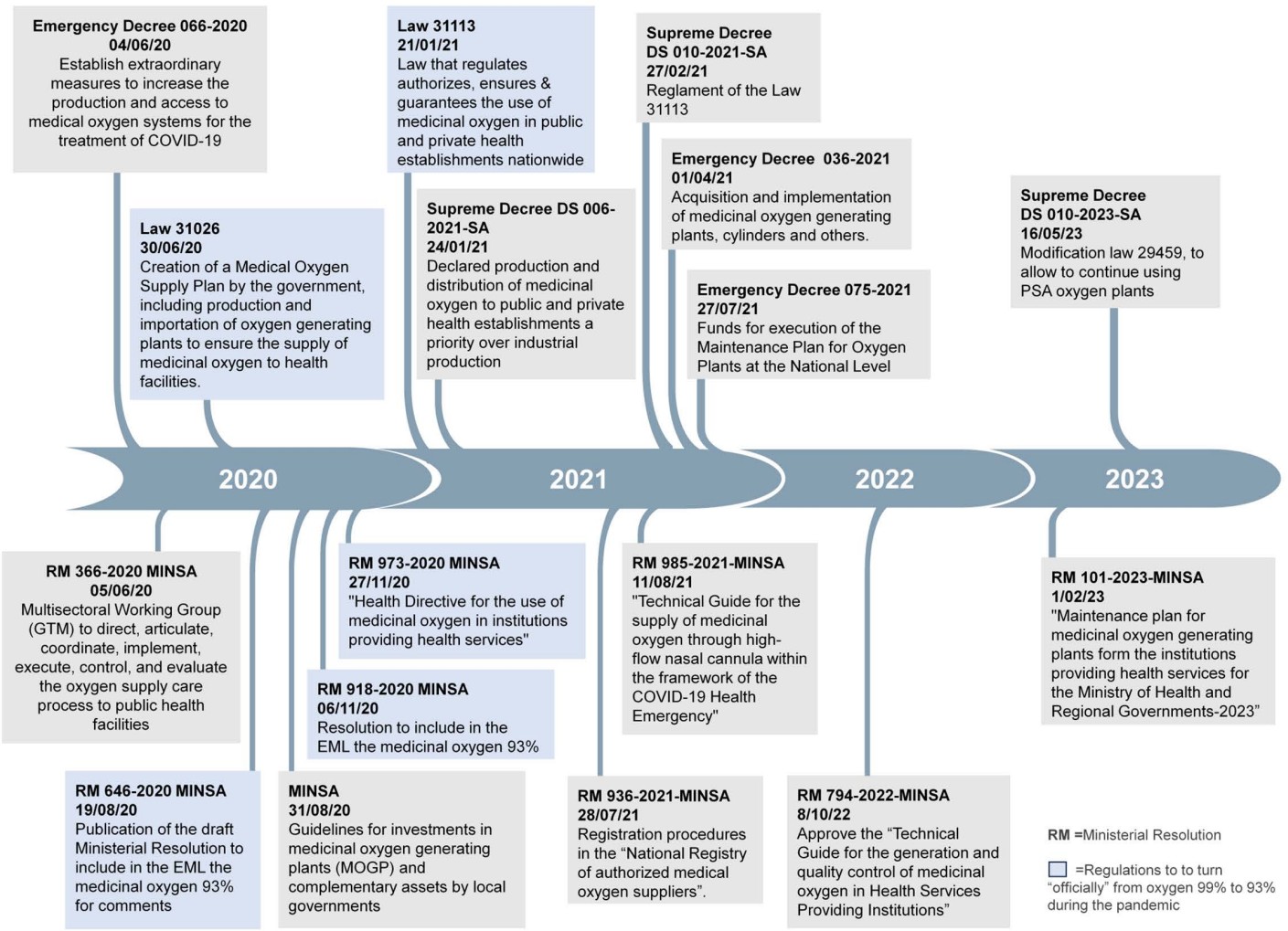

**Fig 2. Timeline of the Regulations regarding Medical Oxygen in Peru during the COVID-19 Pandemic.**

*"... I understand that the cost of a cubic meter of oxygen the real price is 8 soles, 8 to 10 soles and that at the time they sold the cylinder of 10 cubic meters up to 3000 soles, that is, the cubic meter was at 300 soles..."*

*Participant 008*

*"... [In the black market] each cylinder with medicinal oxygen was at 2500, 3500 soles (700 -1000 USD), there are people who have paid up to 7000 soles for one cylinder (2000 USD)".*

*Participant 005*

However, officially lowering the oxygen purity standard to 93% required two laws, three ministerial resolutions, and seven months (Fig 2), highlighting bureaucratic delays potentially linked to economic and political factors as mentioned by the interviewees.

*"... Politics in general are based on interests (...) You didn't even need a law declaring interest (law 31026), you didn't even need a law that creates the National Oxygen Supply System (law 31113), because with the simple act of issuing a ministerial resolution, Ministers could have changed the requirement of oxygen purity from 99 to 93%. Why didn't they do it? Isn't it obvious? because they knew they were going to clash with some (economical) interests, and they preferred not to do it. Or they thought that the pandemic was already over, and they would not need to deal with the issue? Bah...."*

*Participant 005*

For the first time, national guidelines on medical oxygen use and quality control were issued one in 2021 and the other in 2022 [42,43].

Interviewees also suggested irregularities in oxygen procurement, both before and during the pandemic. Before COVID-19, hospitals purchased oxygen directly despite its designation as an essential medicine, which should have been centrally procured by the National Center for the Supply of Strategic Resources in Health from MINSA (CENARES) to reduce costs and improve access. In 2013, however, a ministerial resolution allowed health establishments to make fragmented, direct purchases, —likely intended to simplify processes and expedite access to critical supplies—but this approach increased costs and heightened the risks of inefficiency and corruption [44].

Interviewees from civil society and the business sector observed that authorities delayed in anticipating rising oxygen demand and ignored the need for a coordinated system. When oxygen shortages emerged, a Multisectoral Working Group (GTM) was formed to oversee oxygen supply nationwide [45], but interviewees noted that government response remained slow and ineffective, with insufficient production and a focus on intensive care unit (ICU) bed capacity rather than oxygen access.

*"... The ICU was full, more ICU beds were increasing, they needed more oxygen, that is, a problem was created, a big snowball (...) but the government insisted that we needed more ICU beds and nothing about the oxygen..."*

*Participant 008*

It was evident that it was not possible to address the problem with the local production of oxygen. Eventually, $22 million was allocated to CENARES for oxygen procurement outside the country, but delays and bureaucratic obstacles led to the transfer of these responsibilities to other government agencies. Despite this shift, issues persisted, including problematic purchases such as oxygen cylinders that did not meet Peru's national quality standards.

*"... Everything related to medical oxygen has been very opaque. While it is true that medical oxygen is in national demand (...) it should be bought by CENARES because it is considered a strategic medicine, however, it was bought by the hospitals themselves, due to a change in regulations. Then, when the pandemic came, "Peru Compras" bought it, "Legado" bought it, everyone bought the oxygen, but in the end, they bought and imported oxygen cylinders (...) the cylinders did not have the Peruvian quality standards, they did not have the Peruvian quality standard nanometers …*

*Participant 005*

Among public officials, many cited the absence of centralized leadership, lack of accountability, and high political turnover, which disrupted effective decision-making. Multiple government bodies were involved, but this fragmented structure hampered quick action and solutions. Some interviewees attributed this to personal interests and possible corruption.

Lack of data also worsened the oxygen crisis. Peru lacked accurate data on oxygen demand, production, and distribution, with no inventories of cylinders, LMO tanks and LMO tankers, or oxygen plants in health facilities. Even during the pandemic, demand estimates were based only on case numbers and severity, limiting procurement efficiency.

*"... There was no national oxygen system or notification system, therefore, at the time of the pandemic, our capacity to respond to oxygen needs was not known... What we did was to walk blindly during the pandemic"*

*Participant 005*

The 2021 Oxygen Law (31113) mandated that DIGEMID create a national registry for oxygen suppliers and stock, but this public registry, known as the "National Registry of Medical Oxygen – RENOXI Peru," was not launched until September 2021, eight months later [46]. Although the full database isn't publicly accessible, most information can be found on the REUNIS platform [47], with some data available on a dedicated open-data site called "*oxígeno datos abiertos*" [48].

Interviewees raised concerns about the reliability of this data, pointing out that RENOXI lacks verification mechanisms to ensure data accuracy. In 2023, a cyberattack further compromised RENOXI's registry, resulting in data loss and further diminishing confidence in the system.

*"…I don't know if what RENOXI reports to me (data) is really true… there the consumption is based on the number of cylinders which are delivered from the pharmacy service, if I deliver 3 cylinders of 10 m³ it means that 30 m³ have been consumed, but It is not that they have been consumed, they are just being delivered, consumption is probably lower… or they may forget to register, and nobody supervises or rechecks the numbers"*

*Participant 010*

*"... At the beginning of 2023, the website (RENOXI) went down, due to a computer attack, and the information was lost..."*

*Participant 001*

### Oxygen plants, civil society and private sector contributions

In August 2020, MINSA issued guidelines for local governments on investing in oxygen plants and equipment, allocating funds for the purchase and installation of oxygen plants and cylinders [49]. However, government procurement was slow. Civil society, including academia, NGOs, the private sector outside the oxygen industry, and notably the Catholic Church (with some regional interfaith coalitions), played a crucial role in addressing the government's inability to procure, install and commission oxygen plants. Through fundraising efforts, these groups donated oxygen plants, cylinders, and oxygen

supplies, significantly improving access in areas where government support was insufficient. We found no information of investments or donations related to LMO tanks or tankers for improving LMO transport.

Nationally, *Respira Perú*, an initiative led by the Peruvian Episcopal Conference, the National Society of Industries and other parties, was the most prominent civil society effort, raising funds to purchase and install oxygen supplies and plants. In the regions, local groups organized similar initiatives as COVID-19 cases surged, sometimes with support from political figures and private companies. Many plants obtained through these efforts were donated to hospitals and health establishments managed by MINSA, regional governments, or municipalities. For Respira Perú, a medical council—including representatives from MINSA, EsSalud, and leading doctors—helped decide where to install plants based on need.

However, not all the initiatives were successful. Some interviewees noted that DIGEMID's bureaucratic demands and limited staff hindered plant approvals, leaving oxygen plants stalled at customs without authorization.

> *"…there was a lot of informality during the pandemic… many oxygen generating plants were set up and began to operate without having any type of authorization or anything, because people were simply dying…"*

*Participant 003*

> *"…they [the government officials] put the noose around their necks because they began to demand a requirement, and they did not have officials to carry out the verifications. They issued the documents at the wrong time... we had the oxygen plants at customs, and we could not remove them because there was no authorization from DIGEMID..."*

*Participant 002*

Despite successful oxygen plant installations, operational challenges arose due to high staff turnover, limited training and lack of replacement parts and maintenance plans. By the end of the pandemic, 496 PSA oxygen plants had been installed across the country, funded by national, regional, and private donations.

## The situation of medical oxygen after COVID-19 and lack of preparedness for future pandemics

The process of donating, distributing, installing and commissioning PSA oxygen plants in Peru was marked by opportunistic actions without plans for maintenance or replacement. The allocation was not strategically directed to high-need areas, leading to inequitable distribution, and many plants were left in legal limbo, preventing government investment in their upkeep.

As a result, nearly 200 oxygen plants nationwide are now unused, broken, or abandoned. Since oxygen demand declined post-pandemic, the operation of these plants now depends on local health establishment requirements.

According to one interviewee, MINSA has allocated funds for oxygen plant maintenance since 2021, including for both purchased and donated plants. Yet only part of this budget has been utilized due to delays in legally clearing these assets, which should typically take no more than 90 days.

> *"Up to 3 maintenance plans were made for the oxygen plants: in 2021, 2022 and 2023, and 22 million soles were programmed for it... But I don't think that money has been executed. There are a lot of problems. For example, at the Loayza Hospital there are 3 oxygen plants that were donated, one by the Southern Mining company. The problem is that you can't do anything, you can't invest government funds for maintenance because they are not in someone's name, nothing is formalized. That's a knot..."*

*Participant 011*

Changes in authorities and varying levels of interest or awareness have delayed the legal clearance process. Additionally, some interviewees believe that maintenance delays might be linked to potential corruption in medical oxygen

procurement. Regional barriers include staff lacking adequate training for oxygen plant maintenance and a shortage of qualified service providers. As a result, many health establishments are reverting to pre-pandemic practices, purchasing oxygen from a few primary suppliers instead of maintaining their own plants.

> *"... in Peru, for reasons that need to be investigated, because the truth is that I have heard different versions of the origin, the individual cylinder is used a lot, which is more expensive, more inefficient... but it also lends itself much more easily to dishonest practices (...) I don't know, but what I can say is that today I know that there are hospitals that are ceasing to use oxygen plants to return to the oxygen cylinder system... and they're letting the plant go bad..."*

*Participant 006*

In the 2024 Public Sector Budget Law (Law 31953), about USD 5.7 million has been earmarked for preventive and corrective maintenance for medical oxygen plants, though no clear maintenance plan is in place [50].

When asked about preparedness for future pandemics, most interviewees expressed that despite measures taken during COVID-19—such as providing oxygen plants and concentrators—Peru has not effectively learned from the experience. They believe similar issues would arise in future health crises.

> *"... If we were to face an epidemic again, that kind of circumstances would be very similar, we have not learned the lesson, maybe a little bit or to some degree a better response, but we would still collapse in our need for oxygen."*

*Participant 013*

At a management level, interviewees emphasized the need for a National Oxygen System as a coordinated government-led framework to prepare for emergencies, including robust demand-supply tracking, forecasting and planning, equipment inventories, maintenance, operation supervision, quality control, and negotiates procurement and oxygen distribution across all levels of the health system, to ensure equitable, reliable and sustainable access to medical oxygen-especially during public health emergencies.

> *"... The Executive issued (finally!) an Emergency Decree 066, which says that oxygen drops from 99% to 93% (concentration), that was good. But I want to warn you that this is not the essential issue... The essential thing is to create a National oxygen system, which could strategically assess the demand, assure the supply, create an oxygen quality control system, and understand and assure the equipment, infrastructure, networks, distribution, training etc. etc (...) if we don't have that, it's not going to work..."*

*Participant 005*

Several interviewees support maintaining operational oxygen plants at strategic locations across the country, considering it economically advantageous in the long term. However, some also expressed concerns about the operational and maintenance demands relative to the potential benefits, highlighting the need for further economic studies. All interviewees emphasized the importance of securing local oxygen availability to reduce dependency on external suppliers.

> *"…strategic oxygen plants would surely be a saving for the Government. Nobody has done the math. If you invest in maintenance and operational plants, and adequate well-trained personnel, that is surely a saving and a good investment for the country…"*

*Participant 001*

Addressing future pandemics will also require a stronger focus on human resources. Participants highlighted the need for trained technical personnel to ensure continuous oxygen supply, as well as for dedicated professionals capable of managing critical situations, and improve the training of health professionals on rational oxygen use. They stressed that retaining skilled staff across administrative changes is essential for continuity in crisis response processes.

*" We need medical oxygen in all the country, we need good distribution systems, networks, trained providers and technical personnel. People die due to the lack of oxygen, but we are not counting."*

*Participant 005*

*"... I think that in any case it is necessary to have more technical staff for maintenance of the plant and support, unfortunately the hospital does not have the capacity to be able to hire more people and there are very few people with a good level of qualification for those positions, but it would require personnel because there is a need to have teams that work 24 hours a day and in great demand so you have to have a whole technical support and maintenance staff..."*

*Participant 013*

## Discussion

This study reveals significant barriers to equitable oxygen access in Peru, reflecting broader challenges faced by low- and middle-income countries. We used a problem-driven political economy analysis approach which was well-suited to our study, as the oxygen crisis in Peru was not merely a logistical failure but a symptom of deeper regulatory, market, and governance dysfunctions. This approach allowed us to analyze how fragmented authority, bureaucratic inertia, and political interests hindered timely and equitable access to medical oxygen, while also highlighting how civil society and non-state actors responded in the absence of state coordination. The COVID-19 pandemic exposed Peru's lack of preparedness, stemming from longstanding regulatory and logistical weaknesses. While medical oxygen was designated an essential medicine, the > 99% purity requirement created a "policy trap," limiting competition and concentrating market control within a few large suppliers. The WHO International Pharmacopoeia defines medicinal oxygen as Oxygen 99.5% or Oxygen 93% that contains not less than 90.0% and not more than 96.0% (v/v) of $O_2$, or other products with different oxygen concentrations and/or produced using different production methods, if approved by the appropriate national or regional authority [51]. This regulatory barrier prevented smaller, potentially more affordable providers from entering the market, contributing to severe shortages during the pandemic.

The fragmented and decentralized health system in Peru further complicated oxygen distribution, particularly in rural regions. While the Ministry of Health oversees medical oxygen policies, the lack of a national oxygen strategy and centralized procurement framework led to inconsistencies in supply and an inefficient crisis response. Government entities responded slowly, hindered by bureaucratic delays and limited coordination. Even as COVID-19 cases surged, the focus remained on ICU bed capacity rather than securing sufficient oxygen supplies, highlighting a misalignment in priorities during critical moments.

After the COVID-19 pandemic, many PSA oxygen plants in Peru were left unused or in disrepair due to poor planning, lack of maintenance, limited technical capacity, and unresolved legal status. Nearly 200 donated plants were installed without strategic allocation, and delays in asset formalization prevented their proper use. As a result, many facilities reverted to costlier and less efficient oxygen cylinders. Notably, if a PSA plant remains unused for over two months, the zeolites (nitrogen-adsorbing material) can be damaged—replacing them costs over 50% of the plant's value. With proper maintenance, however, zeolites can last up to seven years [52–54].

Additional barriers for the use of PSA plants reported in the literature include myths about high costs, clinician preference for 99.99% purity over PSA's 93%, and resistance from procurement actors due to reduced informal income

opportunities. A study from India highlights the cost-benefit of using PSA plants over LMO and third-party vendor-refilled cylinders as a source of oxygen. It illustrates that the PSA plants are most economical when they are of higher capacity and used to their maximum capacity, with electricity as the power source [55]. Operational challenges—such as unreliable electricity, sensitivity to heat, humidity, and altitude; lack of trained staff, and limited access to spare parts—also contribute to underuse. Solutions may include technical training, local supply chains, and studies on the clinical equivalence of 93% oxygen, along with alternative energy sources like solar power [21,52].

The experience in Peru during the COVID-19 pandemic highlights a broader issue regarding equipment donations in emergency responses. In Peru, civil society and the private sector played pivotal roles in addressing oxygen shortages by donating oxygen plants and essential supplies. However, the lack of a structured maintenance plan resulted in many of these donated plants falling into disrepair or disuse. This scenario underscores a critical flaw in the global approach to medical equipment donations: while initial donations can provide immediate relief, their long-term effectiveness depends on robust systems for maintenance, training, and integration into local health infrastructure. Without these systems, the impact of such donations diminishes, leaving communities vulnerable when the equipment inevitably degrades or breaks down.

This issue extends beyond Peru, as evidenced during the COVID-19 pandemic when initiatives like the Access to COVID-19 Tools Accelerator (ACT-A) raised over $1 billion, much of which was spent on equipment donations. However, as noted in Unitaid's Global Oxygen Strategic Framework, sustainable planning and investment in medical oxygen systems require far more than initial donations; an estimated $4 billion is needed to develop comprehensive and durable oxygen delivery systems in low- and middle-income countries [56]. Similarly, research published emphasizes the dangers of poorly planned or inappropriate donations, which can overwhelm recipient health systems, lead to waste, or create dependency without ensuring long-term benefits. These examples highlight the need for a paradigm shift in how donations are approached, moving from short-term crisis responses to investments in sustainable systems that empower local health care infrastructure to manage and maintain critical resources independently [57–59].

Our findings underscore the necessity for Peru to establish a National Oxygen System to address the recurring challenges of oxygen shortages and the sustainability of donated equipment. Such a system should integrate supply chain management, maintenance funding, and infrastructure investments to ensure a reliable, locally available oxygen supply. However, the design of a National Oxygen System must also anticipate the risks of bureaucratic inefficiency, political turnover, and inequitable distribution; its effectiveness will depend on transparent governance, defined accountability, and stable financing mechanisms to prevent the recurrence of these same weaknesses. A comprehensive approach would also require training personnel to operate and maintain these systems, reducing dependency on external suppliers and enhancing resilience against future health crises. This proactive strategy would not only address immediate needs but also strengthen Peru's health system capacity to withstand future pandemics or emergencies that demand oxygen availability.

This recommendation aligns with the World Health Organization's recent resolution on strengthening health systems for pandemic preparedness, as highlighted in the document adopted during the 152nd Executive Board Meeting [60]. The resolution emphasizes the need for countries to take strategic actions to ensure access to critical medical resources, including oxygen, as part of broader efforts to achieve universal health coverage and build resilient health systems. It explicitly calls for investments in supply chains, infrastructure, and human resources to address gaps in essential medical supplies, particularly in low- and middle-income countries. Peru's experience during the COVID-19 pandemic serves as a clear example of why these measures are vital, as the absence of a structured system for oxygen supply left many donated plants in disrepair and communities vulnerable.

By establishing a National Oxygen System, Peru would not only meet the immediate recommendations outlined in the WHO resolution but also set a precedent for other countries facing similar challenges. This system would ensure that oxygen resources are not only available but also sustainable in the long term, reducing reliance on ad hoc donations and

external support. Such an initiative would represent a significant step forward in achieving equitable access to essential medical supplies and safeguarding public health in the face of future global health crises.

This study has some potential limitations. First, the relatively small number of key informant interviews—though diverse in stakeholder representation—may not fully reflect perspectives across all regions or levels of the Peruvian health system. Second, the analysis relied heavily on secondary sources such as academic and grey literature and publicly available policy documents, which may be incomplete or biased. Third, the retrospective nature of the study, particularly regarding the COVID-19 response, may have introduced recall bias. We did not explore the role of oxygen concentrators, which were scarce before the pandemic but increased through donations and purchases, particularly benefiting patients with mild to moderate COVID-19. Other oxygen-related devices (e.g., CPAP, BiPAP, ABG machines, and pulse oximeters), which are important components of the oxygen ecosystem, were also outside the scope of this study. Finally, while the political economy analysis offers valuable contextual insights, its findings may not be generalizable to settings with different political, regulatory, or market conditions. Our focus was primarily on oxygen provision for severe cases requiring high-flow or hospital-grade oxygen. In conclusion, addressing the regulatory, logistical, and operational gaps in Peru's oxygen supply system is critical. Developing a coordinated, well-regulated national oxygen framework could prevent similar crises in the future and safeguard public health during emergencies.

## Supporting information

**S1 Appendix. Structured guide for interviews.**
(DOCX)

**S2 Appendix. Academic Search.**
(DOCX)

**S3 Appendix. Prisma Academic Literature.**
(PPTX)

**S4 Appendix. Prisma gray Literature.**
(PPTX)

**S5 Appendix. Prisma gray literature.**
(PPTX)

**S6 Appendix. Data (Transcripts 1–13).**
(PDF)

## Acknowledgments

The authors express their gratitude to the Lancet Global Health Commission on Medical Oxygen Security, Commissioners and Advisors of work package 4 – financing and political economy. Additionally, the authors acknowledge the research librarians at the Karolinska Institute, who conducted literature searches and retrieved articles for this case study as part of the Lancet Global Health Commission on Medical Oxygen Security.

## Author contributions

**Conceptualization:** Patricia J. Garcia, Freddy Eric Kitutu, Freddie Ssengooba, Carina King.

**Data curation:** Patricia J. Garcia, Jehnette M. Guzman, Lizzete Najarro.

**Formal analysis:** Patricia J. Garcia, Lizzete Najarro.

**Funding acquisition:** Freddy Eric Kitutu.

**Investigation:** Patricia J. Garcia, Jehnette M. Guzman, Lizzete Najarro.

**Methodology:** Patricia J. Garcia, Freddy Eric Kitutu, Carina King.

**Project administration:** Patricia J. Garcia.

**Supervision:** Patricia J. Garcia, Freddy Eric Kitutu, Carina King.

**Writing – original draft:** Patricia J. Garcia.

**Writing – review & editing:** Patricia J. Garcia, Freddy Eric Kitutu, Jehnette M. Guzman, Lizzete Najarro, Freddie Ssengooba, Carina King.

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
