## [Decision Letter · Decision Letter 0]

3 Jun 2025

PGPH-D-25-00102

Challenges in the medical oxygen ecosystem of Peru: a political economy analysis

Dear Dr. Garcia,

Thank you for submitting your manuscript to PLOS Global Public Health. After careful consideration, we feel that it has merit but does not fully meet PLOS Global Public Health’s publication criteria as it currently stands. Therefore, we invite you to submit a revised version of the manuscript that addresses the points raised during the review process.

We look forward to receiving your revised manuscript.

Kind regards,

Hassan Haghparast Bidgoli

Academic Editor

Journal Requirements:

    1. Your current Financial Disclosure states, “The case study documented in the article was supported by an award to Makerere University School of Public Health from the Meeting Targets and Maintaining Epidemic Control (EpiC) Project – FHI 360 (prime), made possible by the generous support of the American people through the US Agency for International Development (USAID) contract no. 7200AA19C00002. Cayetano Heredia University, Lima, Peru, provided salary support to PJG, JMG and LN while Makerere University provided salary support to FEK and FS, Every Breath Counts Coalition provided salary support to LG and Karolinska Institute provided salary support to CK.”. However, your funding information on the submission form indicates that you received funding from “FHI 360”. Please indicate by return email the full and correct funding information for your study and confirm the order in which funding contributions should appear. Please be sure to indicate whether the funders played any role in the study design, data collection and analysis, decision to publish, or preparation of the manuscript.

    2. In the online submission form, you indicated that The datasets used and/or analyzed during the current study are available from the corresponding author upon reasonable request. 

3. Uploaded as supplementary information.

Additional Editor Comments :

Both reviewers have requested several clarifications on the findings. Please address these in your manuscript and the response letter. In addition, please address the following:

- Add more details on the problem-based PEA approach and explain why this approach is suitable for your work. In addition, the references used for this (i.e., 17 and 18) approach doesn’t seem to be appropriate. Please add relevant references such as:

 Fritz VB. 2014. Problem-Driven Political Economy Analysis: The World Bank’s Experience. Washington, DC: The World BankHarris D. 2013. Applied political economy analysis: a problem-driven framework. Overseas Development Institute – methods and resource. Overseas Development Institute.

- Line 139: “Each document was reviewed by two independent readers”. I think you mean “two independent reviewers”.

- Please add more details on the study participants/interviewees (n=13). You can add a summary characteristics table, including age/age range, role, gender, years of experience etc.

- Currently the quotations in the results are referenced generically as ‘Health provider’ or ‘national government’. It is common to use participant ID or pseudonym to reference quotations.

- Also, please state the potential limitations of this study at the end of the discussion.

Reviewers' comments:

Reviewer's Responses to Questions

**Comments to the Author**

1. Does this manuscript meet PLOS Global Public Health’s publication criteria?

Reviewer #1: Yes

Reviewer #2: Yes

2. Has the statistical analysis been performed appropriately and rigorously?

Reviewer #1: N/A

Reviewer #2: N/A

3. Have the authors made all data underlying the findings in their manuscript fully available (please refer to the Data Availability Statement at the start of the manuscript PDF file)?

Reviewer #1: Yes

Reviewer #2: Yes

4. Is the manuscript presented in an intelligible fashion and written in standard English?

Reviewer #1: Yes

Reviewer #2: Yes

Reviewer #1: Great paper on an important subject! National oxygen planning is critical, as is equitable distribution to rural areas. It is also complex, political, and hard to do. This paper really captures that well.

A few specific comments below:

1. Can you provide more clarity on Table 1? The "Type of oxygen" column is confusing. It mixes production method, purity level, and distribution method. LOX producers can distribute oxygen in gaseous cylinders or in liquid oxygen tanks. For OxyCusco, Oxigeno Iquitos, Oxygen Loreto, Oxigeno Tao, or Villabolos Rufino, what is the mode of production? 93 or 99% is the purity, but cylinders can have 99% purity. Maybe table should be separated out to show purity level, method of production (LOX, PSA, VSA, etc), distribution method (LOX tanks, cylinders, etc)?

2. (p. 8), 235 Are you sure there was enough oxygen at periphery hospitals? Looks like only Lima levels were established via a Lima source (although I could be mistaken here). Often, levels are low without facilities realizing until health worker training highlights needs.

3. So important to highlight transportation and regional disparities like on line 52 of page 9. Glad this is here.

4. Individually procuring oxygen may have increased cost, but cost may not have been the only driver. Often, individual procurement drastically reduces the time to get emergency supplies. In my experience, often going through national procurement systems can take months or years to procure items that are needed immediately. P. 13 291

5. p.15 402 Suggests that lack of maintenance plan may be due to corruption. Could it also be that most plants were donated without formal maintenance plans? This is a problem we see across the globe. Procuring spare parts across multiple disparate companies is hard. Many hospitals don’t know how to budget for spare parts, because this ongoing costs are not given at the time of installation. It could be corruption, but this is happening globally, and it is partially because donors aren’t planning for 6 months after installation. This suggests lack of foresight and planning more than malice, in my opinion.

6. I see the value of a national oxygen system, but I wonder if there may be a role for the suppliers and/or the private sector. Could existing plants/suppliers partner with the national oxygen system to diversify oxygen sources (less risk of failure), while a national system negotiates more regular delivery and lower price points?

Reviewer #2: Reviewer comments:

• In Financial Disclosure, provide clarity on the following statement, “Every Breath Counts Coalition provided salary support to LG”, seems like an author is missing.

• Line 104: “lower respiratory infections” should be “lower respiratory tract infections”.

• Line 104 to Line 107: If the data (in %) is available for lower respiratory tract infections and non-communicable diseases, it should be stated here like it has been done for chronic respiratory conditions (4.6%).

• Line 201:

o For better understanding, in Table 1, a column should be added which mentions the process/technology used for oxygen manufacturing such as ‘cryogenic distillation’ for LMO for 99% pure oxygen, or ‘PSA/VSA/VPSA’ technology for 93% pure oxygen?

o Also, may be add another column to indicate if the Company is LMO manufacturer only, LMO-based refiller (cylinder refiller), or both.

o Also, it would be great to include to include another column to highlight the form in which oxygen is stored when manufactured. For e.g., large LMO manufacturers (capacity greater than 50 MT/day) will store it in liquid form (99% pure), small LMO manufacturers (4-20 MT/day) would store in gaseous form (99% pure) and PSA/VSA/VPSA technology plants will store it in gaseous form (93% pure).

o Also, if data available, daily manufacturing capacity in case of LMO and PSA/VSA/VPSA manufacturers should be included and if it is an LMO-based refiller, may be their daily refilling capacity could be included.

• Line 226: “rechargeable” should be changed to “refillable”.

• Line 252-253: “Most high-purity liquid oxygen plants are located along the central coast of Peru, while the few PSA plants in the Andean and jungle regions produce only limited quantities of gaseous oxygen at 93% purity”. Are the high purity plants LMO plants and 93% purity plants PSA plants? If yes, it could be specified as shown above.

• Line 254: “High-purity packaging facilities (booster compressor system)”, I believe the author is talking about facility where liquid medical oxygen is packed into gaseous cylinders using a booster compressor system. If yes, please include ‘booster compressor system’ in bracket as shown above.

• Line 257: Is the author talking about lack of LMO tankers to move LMO from manufacturing sites to hospitals with LMO tanks and LMO-based refillers for cylinder refilling? If yes, please clearly specify that. Or were there also not enough normal trucks/vehicles to move the 1500 L and 7000 L gaseous cylinders?

• Line 323: “isotanks” should be “LMO tanks and LMO tankers”.

• Line 355: “procure and implement” should be “procure, install and commission”.

• Line 381: “496 oxygen plants” should be “496 PSA oxygen plants”.

• Line 385: “donating, distributing, and implementing oxygen plants” could be changed to “donating, distributing, installing and commissioning PSA oxygen plants”.

• Line 469: “< >” should be changed to “>”.

• Line 548: “Every Breath Counts Coalition provided salary support to LG”, may have to be removed.

• Between Lines 385 and 423, if the authors feel appropriate, they could add the following points about why PSA plants are not being used as much:

o Myths about PSA plants being too expensive. There is literature on it from India.

o Clinicians not preferring PSA plants 93% oxygen as they feel 99.99% purity is what is needed, may be studies comparing clinical outcomes of 93% pure oxygen vs 99% pure oxygen should be conducted to address this concern.

o PSA plants remove dependence on third party vendors which impacts under the table income of procurement managers.

o PSA plants operations and maintenance need more technically experienced staff which is not available.

o PSA plants need continuous electricity, voltage stabilisers and diesel generator set for back up (which can be very expensive to run depending on the diesel prices in the country). Any evidence on using solar power source to provide uninterrupted electricity for their operations?

o PSA plants are sensitive to heat, humidity, and altitude, which affect the purity.

o PSA plants donated from overseas have no local supply chain of spare parts and once they breakdown, they become defunct as there is no local spare parts industry to fill the gaps.

Moreover, it should be highlighted that if PSA plants are left unused for over 2 months, the zeolites (nitrogen adsorbing material) get damaged and replacing them alone costs over 50% of the PSA plant procurement cost. If PSA plant is maintained well, the zeolites should last up to 7 years without issues. Therefore, if the demand is low and they are not required to be used, they should be switched on once every 2-4 weeks and run for about 8-10 hours and then switched off, and this should be repeated until they are used more regularly. To increase their utilisation, booster compressors could also be used to fill gaseous cylinders which could be supplied to nearby facilities (in rural areas) in hub-spoke model. However, to install booster compressors on hospital premises may require necessary regulatory approvals from safety organisations as pressures for refilling are as high as 250 bar which can be lethal if done without training and supervision. Are there regulations for PSA plants being used for refilling cylinders with booster compressor systems?

The authors may also talk about a new French technology for on-site oxygen production, the ION Ionic Oxygen Generator, has been developed by NOVAIR in collaboration with NASA. This generator produces ultra-pure oxygen directly from ambient air, offering a reliable, compact, and user-friendly alternative to traditional oxygen supply methods. The ION generator can produce oxygen with purity levels of greater than 99.99%.

Some other points the authors can consider including to improve the understanding of the reader:

• Talk about availability of LMO tanks at the facilities and LMO tankers to move the liquid oxygen from manufacturing site to the hospitals with LMO tanks or LMO based refillers. Any new installation of LMO tanks at hospitals during or post COVID-19? We see that 496 PSA plants were installed. And how about procurement of any new LMO tankers to move the liquid oxygen from one place to another?

• Any information of any liquid argon or nitrogen tankers being converted to LMO tankers to meet the demand?

• Any information on converting industrial gaseous oxygen cylinders to medical oxygen gaseous oxygen cylinders?

• Any use of IoT devices to monitor the consumption of oxygen from LMO tanks to notify the LMO supplier on refilling?

• Any use of IoT devices to monitor the utilisation of PSA plants?

• Role of portable oxygen concentrators for home-care use and hospital use (especially in secondary and primary care facilities) have not been discussed. Are they not available or used in Peru?

• How did the country fair on availability of oxygen concentrators, ventilators, HFNC, CPAP and BiPAP devices? I believe there should not have been any shortage of patient monitors, ABG machines and various types of pulse oximeters.

• Are all the hospitals equipped with medical gas pipelines systems and manifolds? This a must have for PSA plants and LMO tanks.

• Was there no black marketing or hoarding of oxygen cylinders or concentrators?

• Was industrial supply redirected for medical use during COVID-19?

**Do you want your identity to be public for this peer review?** For information about this choice, including consent withdrawal, please see our Privacy Policy

Reviewer #1: **Yes: ** Victoria Smith

Reviewer #2: **Yes: ** Dr Varun Manhas

---

## [Decision Letter · Decision Letter 1]

29 Sep 2025

PGPH-D-25-00102R1

Challenges in the medical oxygen ecosystem of Peru: a political economy analysis

Dear Dr. Garcia,

Thank you for submitting your manuscript to PLOS Global Public Health. After careful consideration, we feel that it has merit but does not fully meet PLOS Global Public Health’s publication criteria as it currently stands. Therefore, we invite you to submit a revised version of the manuscript that addresses the points raised during the review process.

We look forward to receiving your revised manuscript.

Kind regards,

Hassan Haghparast Bidgoli

Academic Editor

Journal Requirements:

Additional Editor Comments:

Thanks for revising the manuscript and addressing the reviewers' comments. However, reviewers still have some concerns regarding the responses and the revised manuscript (see below). Please address them in your response and updated version of the manuscript.

Reviewers' comments:

Reviewer's Responses to Questions

**Comments to the Author**

Reviewer #1: All comments have been addressed

Reviewer #2: All comments have been addressed

publication criteria?

Reviewer #1: Yes

Reviewer #2: Yes

3. Has the statistical analysis been performed appropriately and rigorously?

Reviewer #1: N/A

Reviewer #2: N/A

4. Have the authors made all data underlying the findings in their manuscript fully available (please refer to the Data Availability Statement at the start of the manuscript PDF file)?

Reviewer #1: Yes

Reviewer #2: Yes

5. Is the manuscript presented in an intelligible fashion and written in standard English?

Reviewer #1: Yes

Reviewer #2: Yes

Reviewer #1: The revisions look good. Very glad to see more of the oxygen system clarified. Oxygen regulations are an important part of the system and have a huge impact on availability. The problems of national oxygen distribution are also key, and more literature is needed on these topics.

Two small proof reading comments:

p. 11, Line 314 doesn't quite make sense

p. 14 &15, Some exact sentences repeat in different paragraphs "Regional barriers...maintaining plants."

A question I still have: While a National Oxygen System would certainly help with lack of data and clear regulations, how can it be ensured that a National Medical Oxygen System would avoid the same pitfalls of long approval times, national politics, and lack of distribution to politically less-important geographies if it is a government agency? Would it have the power to ensure budget? Are there guidelines or specific by-laws that would ensure efficient and reliable access, so that parallel private oxygen acquisition systems are not needed and do not emerge? Some of this may be outside the scope of this text, but could be great future research.

Reviewer #2: Line 66: “lower respiratory infections” should be “lower respiratory tract infections”.

No further review is needed from my end once this final comment is addressed.

**Do you want your identity to be public for this peer review?** For information about this choice, including consent withdrawal, please see our Privacy Policy

Reviewer #1: **Yes: ** Victoria Smith

Reviewer #2: **Yes: ** Dr Varun Manhas

---

## [Editor Report · Decision Letter 2]

19 Nov 2025

PGPH-D-25-00102R2

Challenges in the medical oxygen ecosystem of Peru: a political economy analysis

Dear Dr. Garcia,

Thank you for submitting your manuscript to PLOS Global Public Health. After careful consideration, we feel that it has merit but does not fully meet PLOS Global Public Health’s publication criteria as it currently stands. Therefore, we invite you to submit a revised version of the manuscript that addresses the points raised during the review process.

We look forward to receiving your revised manuscript.

Kind regards,

Hassan Haghparast Bidgoli

Academic Editor

Journal Requirements:

1. Please amend your detailed online Financial Disclosure statement. This is published with the article. It must therefore be completed in full sentences and contain the exact wording you wish to be published.

a) State the initials, alongside each funding source, of each author to receive each grant. For example: “This work was supported by the National Institutes of Health (####### to AM; ###### to CJ) and the National Science Foundation (###### to AM).”

For more information, please go to our submission guidelines:

https://journals.plos.org/globalpublichealth/s/submission-guidelines#loc-financial-disclosure-statement

2. Please ensure that the funders and grant numbers match between the Financial Disclosure field and the Funding Information tab in your submission form. Note that the funders must be provided in the same order in both places as well.

Additional Editor Comments:

Thank you for addressing reviewers concerns. Both reviewers have some additional minor comments. Please address them in your manuscript.
---

## [Editor Report · Decision Letter 3]

3 Dec 2025

Challenges in the medical oxygen ecosystem of Peru: a political economy analysis

PGPH-D-25-00102R3

Dear Dr. Garcia,

We are pleased to inform you that your manuscript 'Challenges in the medical oxygen ecosystem of Peru: a political economy analysis' has been provisionally accepted for publication in PLOS Global Public Health.

Best regards,

Hassan Haghparast Bidgoli

Academic Editor
